# Socioeconomic Inequalities in the Prevalence of Non-Communicable Diseases among Older Adults in India

**DOI:** 10.3390/geriatrics7060137

**Published:** 2022-12-05

**Authors:** Bikash Khura, Parimala Mohanty, Lipilekha Patnaik, Keerti Bhusan Pradhan, Jagdish Khubchandani, Bijaya Kumar Padhi

**Affiliations:** 1International Institute for Population Sciences, Mumbai 400088, India; 2Department of Community Medicine, Institute of Medical Sciences & SUM Hospital, Siksha ‘O’ Anusandhan (Deemed to be University), Bhubaneswar 751030, India; 3Department of Public Health Management, Chitkara University, Rajpura 140401, India; 4Department of Public Health, New Mexico State University, Las Cruces, NM 88003, USA; 5Department of Community Medicine and School of Public Health, Postgraduate Institute of Medical Education and Research, Chandigarh 160012, India

**Keywords:** non-communicable diseases, hypertension, diabetes, inequality, disparity, preventive care, India

## Abstract

Understanding socioeconomic inequalities in non-communicable disease prevalence and preventive care usage can help design effective action plans for health equality programs among India’s aging population. Hypertension (HTN) and diabetes mellitus (DM) are frequently used as model non-communicable diseases for research and policy purposes as these two are the most prevalent NCDs in India and are the leading causes of mortality. For this investigation, data on 31,464 older persons (aged 60 years and above) who took part in the Longitudinal Ageing Survey of India (LASI: 2017–2018) were analyzed. The concentration index was used to assess socioeconomic inequality whereas relative inequalities indices were used to compare HTN, DM, and preventive care usage between the different groups of individuals based on socioeconomic status. The study reveals that wealthy older adults in India had a higher frequency of HTN and DM than the poor elderly. Significant differences in the usage of preventive care, such as blood pressure/blood glucose monitoring, were found among people with HTN or DM. Furthermore, economic position, education, type of work, and residential status were identified as important factors for monitoring inequalities in access to preventive care for HTN and DM. Disparities in non-communicable diseases can be both a cause and an effect of inequality across social strata in India.

## 1. Introduction

The age distribution of the population has changed dramatically across the globe. Declining mortality and advancements in medical and scientific knowledge are the drivers of prolonged life expectancy [1]. By 2050, it is estimated that more than 200 million people over the age of 60 would be added to the global population [2]. Specifically, the population of low and middle-income nations, such as India, is rapidly aging. For example, in India, the proportion of older adults has increased from 5% of the total population in 1951 to 8.6% in 2011 and is expected to reach 20% of the overall population by 2050 [3]. Population aging is a global concern regarding labor markets, health system requirements, economics, and social welfare. As people begin to age, they are more likely to develop various chronic diseases. Not only do older people have to deal with a plethora of chronic diseases, but there are also differences in health status between sub-age groups of older people, with some groups faring better than others based on a variety of socioeconomic factors. This makes access to health care even more unjust and unequal [4]. Despite progress in improving access to health care, disparities based on sex, caste, wealth, education, and residence prevail [5]. To achieve health equity, policymakers must measure inequalities in health care, especially as it relates to wealth and income inequalities [6].

Studies show that non-communicable diseases (NCDs) that differently manifest between the wealthy and the deprived are major predictors of premature mortality and differences in life expectancy among older people [7,8]. According to the Global Burden of Diseases (GBD) report, cardiovascular diseases, chronic lung illness, and DM are the leading causes of death globally [1,9,10]. In India, NCDs claim the lives of nearly 6 million people [11]. The GBD collaborators of India observed that primarily, cardiovascular diseases contributed to more than a third (34.3%) of India’s mortality burden [12] and diabetes mellitus accounts for 2.5% of the total burden of mortality [13]. In addition, the prevalence of NCDs continues to escalate in India with major public health implications, predominantly in older adults.

In terms of disease burden, NCDs account for 62 percent of DALYs jeopardizing long-term growth and affecting individual productivity in India [14]. NCDs account for 40% of hospital admissions and 35% of outpatient admissions, causing loss of productive years of life. This, in turn, creates further income inequalities by loss of income and wealth in households where the poor are disproportionately affected [15]. Furthermore, in nearly half of the NCD cases, the majority of the cost is shifted to individuals and families (i.e., out-of-pocket payments) [16].

It is well established in the scientific literature that socioeconomic status is the most important factor influencing the distribution of chronic diseases in and within communities [17,18]. Globally, the relationship between the occurrence of NCDs and economic status varies widely with pronounced regional differences [19]. The socioeconomic pattern of inequality in the prevalence of NCD differs from one country to another and within nations depending on prevalent health and social policies [20]. For example, studies from India itself have found an association between the prevalence of NCDs such as hypertension(HTN) and diabetes mellitus (DM) and the socioeconomic position of individuals [21,22,23,24].

In addition to NCDs, there are differences based on SES in the utilization of preventive therapies and adherence to medications for NCDs [25,26,27]. Diabetic patients must be monitored for and advised on blood glucose checks, adherence to hypoglycemic medicines, and dietary habits to regulate blood glucose and avoid complications [28]. Despite the relatively modest expenditure for preventative care for NCDs in most nations, people from economically underprivileged sections do not often utilize or are offered preventive care and are as a consequence, more likely to experience premature mortality due to chronic diseases than their wealthier counterparts [20].

India, with its diverse socio-cultural, economic, and geographic makeup has recognized and responded, by implementing multi-sectoral NCD prevention initiatives to tackle health inequalities [29,30]. For example, policy and attention from governance, such as the implementation of national policies in line with the SDG paradigm [16] have remained the mainstay in addressing the NCDs. Specific initiatives from the government and community agencies have focused on factors such as prevention, education, treatment, and rehabilitative services to achieve the SDG goal of 3.4 (which suggests a reduction by a third in early deaths caused by NCDs by the year 2030 [31]. However, it has not been thoroughly explored whether chronic disease prevention techniques and initiatives were widely implemented and equitable in India. Very little information is available about socioeconomic inequality in the prevalence of NCDs, especially among older adults. So, such a study is much needed because NCDs (HTN and DM) vary among members of a given society and have a remarkable influence on the well-being of India’s aging population. Thus, the purpose of this study was to comprehensively and systematically examine socioeconomic inequality in the prevalence of NCDs and the use of preventive care among the aging population and also, to explore the role of other socioeconomic predictors in health inequalities for older Indian adults.

## 2. Materials and Methods

### 2.1. Data

The analysis is based on data collected from the 2017–2018 national survey of longitudinal aging in India (LASI), which focuses on the well-being, economy, and social elements of India’s aging population [32]. The International Institute for Population Sciences (IIPS) in Mumbai conducted the study in partnership with international collaboration [33]. The sample consists of 72,250 older citizens who are 45 years of age or older along with their spouses from across India except for Sikkim state. The final units of observation were chosen by LASI using a multistage stratified area probability cluster sampling design. In addition, the LASI 2017–2018 report provides details of the sampling frame and methodology [32]. This survey on aging provides empirical evidence on health, demographics, and, social determinants. The sizes of the LASI datasets that were finally included in our analyses are described in Figure 1.

### 2.2. Variable Description

In this study, the eligibility criteria of the study population was older adults aged 60 years and above. We used data from 31,464 people (15,098 older males and 16,366 older females). Further adults below 60 years were excluded from the analysis. The following key variables were considered for analysis.

Outcome variable: The presence of hypertension (HTN) and diabetes mellitus (DM) along with its preventive treatment (use of medicine) among older adults with HTN and DM. In this study, we used self-reported HTN and DM as model NCDs as they are the most common NCDs in India and the leading causes of mortality [24,34,35].

Self-reported HTN /DM (diagnosed by a physician) was used to determine the existence of HTN/DM by answering questions such as“Have you ever been diagnosed with hypertension?” along with “Have you ever been diagnosed with diabetes?”. All outcome variables were binary. The existence of HTN and/or DM was the first group of variables.

The second group of variables was: If they had HTN or DM, whether they took their medicine as prescribed by their doctor, and if their blood pressure or blood glucose had been checked. Only those who reported having HTN or DM were investigated further on their medication use.

Exposure variable: As suggested by previous studies [36], in this study, house consumption spending per equivalent adult was also used as a proxy indicator of economic status. Further economic status was used in regression analysis, i.e., monthly per capita consumption expenditure (mpce_quintile) was divided into quintiles: poorest (lowest 22%, *n* = 6484 MPCE), poorer (lower 22%, *n* = 6477), middle (middle 21%, *n* = 6416), richer (higher 19%, *n* = 6172) and richest (highest 16%, *n* = 5917).

Explanatory variables:

The explanatory variable was categorized as participants of the 60–69 years age group, 70–79 years age group, and 80+ years age group. Gender was a binary variable (i.e men and women). Religions were categorized into four groups first Hindu, second Muslim, third Christian, and the fourth group as Others. Similarly, caste groups were categorized as scheduled caste (SC), scheduled tribes (ST), other backward class (OBC), and the fourth group as Others.

The other variable was classified as educational attainment as ‘illiterate’, ‘Primary or below’, ‘Secondary’ and ‘college and above’. Employment status was taken as ‘Unemployed’ and ‘Employed’. Marital status was taken as ‘Married’ or ‘Single’.

A direct acyclic graph (DAG) was used to identify covariates that should be adjusted while developing the model. Figure 2 shows a simplified DAG to facilitate readability. Covariates and arrows that have no bearing on the analysis are excluded.

### 2.3. Statistical Analysis

For statistical analysis a descriptive, basic weighted point estimate, “relative inequality index” (“RII”), “economic-related concentration index” (‘C’), and “decomposition analysis” for ‘C’ were undertaken in the study. In previous studies, “RII” and “C” are commonly used to estimate health or healthcare inequalities [37,38]. All estimates were calculated using survey weights and taking the complex survey design into account. The missing data were excluded from the analysis. Analysis after the exclusion of cases with missing data did not change the estimates.

The RII index compares extremes, whereas the C index measures socioeconomic inequality across the entire socioeconomic spectrum. For both HTN and DM, analyses were conducted sequentially.

#### 2.3.1. The “Relative Index of Inequality”

“RII” indicators are used to compare ailment prevalence rates between those with the most and least socioeconomic status positioned people.” It is mainly utilized in epidemiology and public health [37].

The predicted prevalence of disease ratio between the disadvantaged and wealthy groups can be read as “RII”. As a result, an RII value of more than 1 indicates a greater prevalence among the poorest members of society and the other way around.

As proposed in other studies, Poisson regressions, with robust error variance, were used between economic groups to estimate the prevalence rate ratio adjusted for confounders (i.e., ‘RII’), as the outcome was binary in this study [39]. The ‘RII’ were calculated in 2 stages: they were first corrected for age and sex and then secondly, they were adjusted for education, occupation, and other social and economic characteristics.

#### 2.3.2. Concentration Index

‘C’ assesses disparities across the socioeconomic spectrum and accounts for socioeconomic inequalities in health conditions/preventive care [38]. The “C” index has a value ranging from −1 to 1, with 0 indicating a perfect equality condition. So, a +ve C indicates that health condition/ preventive care is to a greater extent concentrated in the wealthier population and the other way round.

The C is denoted in the formula as:(1)$$“C”=\frac{2}{\mu }\mathrm{cov}\left(y_{i}r_{i}\right)$$
where y denotes the health/health care factor (in this case, the prevalence of HTN/DM and its preventive treatment), μ denotes the mean of the health and healthcare variable, and $r_{i}$ denotes individual fractional ranking from a stretch of 0 to 1 in the economic distribution of people.

#### 2.3.3. “Decomposition Analysis” for “C”

The “C” can be divided into demographic and socioeconomic components, according to Wagstaff et al. The contribution to that component is determined by the product of the degree of economic inequality and the sensitivity of the outcome variable [40].

As the outcome variable is binary; hence, a probit regression model has been used to estimate the partial effect of each predictor variable. However, the findings are not interpreted to imply a causal relationship [41]. The outcome variable (y) is modeled using the following model.
(2)$$y_{i}=\sum_{k}\beta_{k}x_{\mathrm{ki}}+\varepsilon_{i}$$

Here “βk” denotes the number of partial effects. Each regressor dy/dx is assessed as the mean of the sample, and “ε “as the error term from the mathematical model. “x_k_” denotes a set ofpredictors.

The ‘concentration index C” (y) can be decomposed as follows:(3)$$“C”=\left(\frac{\beta_{k}{\bar{x}}_{k}}{\mu }\right)C_{k}+\frac{{\mathrm{GC}}_{\varepsilon }}{\mu }$$

Here “β_k_” depicts the k regressors’ partial effects (that is, predictor), as calculated in Equation (2).

Each regressor’s mean is $\bar{x}$_k_ and μ is taken as the mean value of the body condition/ preventive approach. “C_k_” refers to the “regressor concentration index”, and “GC_ε_” refers tothe “generalized concentration index of ε”.

Further inequality is represented as the residual component $\left(\frac{{\mathrm{GC}}_{\varepsilon }}{\mu }\right)$ that the regressors don’t explain. The deterministic element $\left(\sum_{k}(\frac{\beta_{k}{\bar{x}}_{k}}{\mu })C_{k}\right)$ concentrates on two components. The two components are the health elasticity and (C_k_), which is the degree of unequal distribution of distinct regressors throughout the economic spectrum in relation to regressor $\left(\eta_{k}=\beta_{k}\frac{{\bar{x}}_{k}}{\mu }\right)$.

Further, each regressor’s absolute contribution of each regressor was calculated (Q_k_ = η_k_C_k_). Each regressor can contribute in either a positive or a negative way.

According to Equation (1), even if a predictor variable has a large effect on the health condition/ preventive care factor, when the factor is uniformly distributed in the rich section and poor, the predictor variable may not have a significant contribution to inequality [as per Equation (3)].

Moreover, disparities in health condition/preventive care are related to demographic variables such as age, as well as socioeconomic variables and urban-rural indicators. To determine whether demographic characteristics are unavoidable for modification, policymakers can choose to focus on socioeconomic inequality [42]. The age and gender-adjusted “C” was computed in the study by removing the contributions of age and gender from the overall “C” using the decomposition results [42,43].

The sampling methods were adjusted using national-level (indiahhweight) sampling weights to establish that the findings accurately represent the Indian elderly section of the population. The STATA14.2′s Svyset function was used to create all models. Furthermore, the standard errors for each of the analytic models were adjusted for clustering at the family level [44].

Ethical approval: Necessary requisite ethical approval has been obtained from the International Institute for Population Sciences (IIPS) team involved in the data collection method.

## 3. Results

The data for the distribution of descriptive variables used in the study are shown in Table 1. HTN and DM were prevalent in 33% and 14% of the population, respectively. Among hypertensive older adults, 77% were taking medications. Similarly, among people suffering from diabetes, 82% were taking medications.

The prevalence of HTN and DM, as well as the proportion of people who receive a preventive care service (treatment seek) with that of the economic quintile, the RII, and the economically linked ‘C’, is seen in Table 2. The socioeconomic-related “C” measured disparities across the economic spectrum, whereas the “RII” measured differences between the poorest and highest economic sections. The prevalence ratios for HTN and DM were 0.18 (95% CI: 0.17–0.20) and 0.06 (95% CI: 0.05–0.06) when RIIs were adjusted for various socioeconomic characteristics. The “Cs” with +ve values indicate that HTN and DM are more common in economically advantaged groups (HTN: C = 0.13; 95% CI, 0.09–0.16; DM: C = 0.10; 95% CI, 0.06–0.14). The adjusted prevalence ratio for individuals with HTN and DM who used appropriate treatment to manage high blood pressure was 0.56 (95% CI 0.53–6.60), and the adjusted prevalence ratio for blood sugar was 0.69. (95 percent CI 0.60–0.74). Positive “Cs” indicates that the wealthy are favored in preventive care for HTN and DM patients.

### The ‘Cs’ Have Been Decomposed

The results of “Cs” for HTN and DM, as well as patient preventative care, are presented in Table 3.

The appearance of HTN and DM among those with preventive care (including those taking medication) was significantly and positively related to economic status, as seen by the partial impact estimates. The partial effect estimates on the presence of HTN for the poorer was 0.035, for the middle was 0.054, for the richer was 0.083, and for the richest was 0.104 were significant. Respondents in higher socioeconomic levels were likely to have HTN or DM. Those with HTN or DM were more likely to practice more effective preventive care strategies. The level of education was significant for HTN prevalence and use of preventive care; however, the level of education for preventive care of DM was inconsistent. Employment status was significantly and negatively associated with the prevalence of HTN and DM and their prevention. Respondents with HTN or DM who lived in cities were more likely to take their medications as prescribed.

We presented a detailed decomposition of the HTN prevalence using the ‘employment’ variable. The “Employed” group contributed 0.003 to the overall C. Further, the percentage contribution (2.45%) is calculated by dividing the absolute contribution (0.003) by the HTN prevalence concentration index.

Economic status, employment, caste, and educational level were the four main contributors to inequalities in NCDs (i.e., based on the contributions of different socioeconomic factors to differences in the prevalence of HTN or DM). Furthermore, economic position, caste, education, urban-rural location, religion, marital status, and occupation all played key roles in disparities favoring wealthy older adults in the prevention of HTN or DM. Socioeconomic status was found to have a significant influence on the prevalence of HTN and DM, as well as disparities in preventative treatment favoring wealthy older adults in India. The contribution of economic status was determined by its significant influence on the prevalence of the disease condition and associated preventive care, as well as their unequal distribution.

Table 4 summarizes the impact of “Cs” and age, as well as adjusted for sex “Cs,” on the occurrence of HTN and DM, as well as preventative therapy. The age and sex group contributions were estimated by combining the contributions of the age plus sex dummy variables using similar approaches as the previous group.

Figure 3 shows the distribution of the concentration curve of the cumulative proportion of the presence of HTN and the use of HTN medicine use, against the cumulative percentage of wealth.

Figure 4 shows the distribution of the concentration curve for the cumulative proportion of the presence of DM and the use of DM medicine use, against the cumulative percentage of wealth, in India (LASI).

Table 4 indicates economic status as the most important sociological factor (41–47%), while other sociological factors were less important (2–16%). Total observed discrepancies were influenced less by age and sex. For the prevalence of HTN and DM, the age-sex-adjusted Cs were 0.135 and 0.10, respectively. Taking into account both gender and age, the wealthy still had a higher prevalence of HTN and DM than the impoverished. The adjusted ‘C’ for age and sex to use medications for HTN and DM were 0.102 and 0.093, respectively. Even when gender and age differences were taken into account, there remained substantial inequality favoring the wealthy in preventive treatment for people with HTN or DM.

## 4. Discussion

In this large, national, comprehensive, and first-of-its-kind study of older adults in India, we found socioeconomic variations in the occurrence of non-communicable diseases and their preventive management as well as quantified the role of predictive variables in these disparities. Wealthy older adults in India have a higher prevalence of HTN and DM than the poverty-stricken section. There was a clear inequality in preventive care, and the wealthy benefited from adequate medication use and blood pressure/glucose monitoring among people with HTN or DM. The inequality in the prevalence of HTN and DM, as well as preventive care, was primarily driven by socioeconomic factors, and unobserved associations were revealed using decomposition analysis. For example, the prevalence of HTN and DM is accumulated among wealthy older adults. These results are in agreement with the findings from different parts of India [23,24,43]. A large number of studies conducted in low- and middle-income nations have shown a significant association between economic gradients and health outcomes [44,45,46,47]. Some studies in India also show an inverse economic gradient [48,49]. However, such a pattern is mostly observed in high-income countries [50,51,52]. In our analysis, the most likely explanation is that older people of lower socioeconomic statuses were unable to receive or were not offered therapy for HTN and DM. Some studies have also shown that the association between socioeconomic status and the prevalence of HTN and DM may vary or even reverse over time, especially in economically developing countries [17,53]. Currently, higher socioeconomic classes in India are at higher risk of HTN and DM as a result of rapid economic growth and a westernized lifestyle [54,55,56].

Differences in social, cultural, economic, and healthcare systems contribute to the uneven distribution of NCDs [20]. As a result of socioeconomic gaps in the prevalence of HTN and DM, our analysis also provides insights into the prevalence of chronic diseases among India’s older population. Strengthening preventative care for older adults with HTN or DM reduces complications and increases the likelihood of survival and well-being. However, our findings suggest that among older people with HTN or DM, there are explicit differences in preventive treatment that benefit the wealthy segment of society. These results corroborate earlier research showing an association between lower SES levels is associated with worse health outcomes among patients with NCDs around the world [20,57]. Specifically, cost, access, and quality of care all vary by SES in countries such as India.

Our findings highlight several critical socioeconomic factors related to preventive care for older people with HTN or DM. In this study, those in the poorest, middle, and richest socioeconomic sections were more likely to experience NCD. Previous studies have found that economic position, education, work status, and urban-rural divide have all been identified as key socioeconomic factors related to the use of preventive care services [25,58]. People’s health-seeking attitudes are often influenced by a variety of sociocultural factors that are linked to their socioeconomic level. This can be critical in deciding whether or not to pursue an NCDs prevention strategy [59]. The widespread diabetes burden is becoming a public health problem in both developed world and developing countries, with researchers estimating that one out of every two elderlyis likely to be diabetic or prediabetic [60]. Furthermore, significant socioeconomic inequalities in hypertension exist in developing countries such as Kenya, which are mostly explained by metabolic risk factors (BMI), individual health behaviors, and socioeconomic factors [61]. Interestingly, in developed countries, socioeconomic status, particularly wealth, is a substantial independent predictor of incident hypertension [62]. However, the current study found that the older population had the greatest concerns, with economic position, caste, educational level, urban-rural residency, religion, marital status, and work status all playing important roles in disparities favoring the richest section of society in the prevention of HTN and DM. Likewise, another study conducted inIndia found that the highest SES regions had the highest prevalence of hypertension, whereas the lowest SES regions had the lowest. In general, the study reflected higher income was related to a higher prevalence of hypertension. On the contrary, there was no evidence that educational attainment altered the relationship between income and hypertension [63]. Furthermore, in this study, after controlling for age and gender, we observed that the wealthy were more likely than the poor to have HTN and DM.

The U.N Sustainable Development Goals have emphasized the importance of paying more attention to elements related to the impact of socioeconomic factors on health inequalities (e.g., education, participation in policymaking, employment, etc.). As a result, targeted efforts should be made to address the disparities in the burden of non-communicable diseases (NCDs) and their preventive care services among the older population (which vary by socioeconomic status, employment, caste, and level of education, as evidenced by the findings of the study). Income and social support, health education and awareness campaigns, programs mimicking universal health coverage schemes for the marginalized, creating a pipeline of community health workers focusing on geriatric care, and creating a culture of NCD prevention and management for the elderly in primary and community care systems would further help reduce the disparities.

There are certain potential limitations of our investigation. The converse interpretation is also plausible because the current study was a cross-sectional study design with no full analysis of the link between causal factors of the prevalence of HTN or DM and its preventive care services. Furthermore, data on health outcomes were obtained by self-reporting, which may have led to an overestimation of the association between socioeconomic status and the prevalence of non-communicable diseases, and the utilization of preventive care. However, the data were the best available for analyzing potential associations between socioeconomic status and NCDs among the older population. In addition, the survey employs a complex sampling technique to obtain a representative sample at a low cost. Furthermore, using “C” for binary health variables captures the extent of socioeconomic inequality over the full socioeconomic range rather than just the variable’s upper boundary. Finally, in questionnaire-based surveys, recall biases are unavoidable, particularly for health conditions that occurred months before the survey.

Despite these limitations, ours is the largest investigation of NCDs in elderly Indians and the disparities in disease burden and lack of preventive care utilization. In addition, our utilization of a nationwide large survey dataset helped identify key socioeconomic differences in preventive treatment for patients in older adults in India. Future studies should consider tracking changes in non-communicable disease prevalence and inequalities in its prevention, as well as evaluating the effectiveness of healthcare programs in lowering these inequalities. Unlike previous studies on NCDs, the study’s findings distinctly demonstrated how socioeconomic factors and NCDs-related factors are linked to inequality among older people. Given the high prevalence of NCDs among older people and the growing aging population, the findings of the research emphasize the importance of strengthening health care, preventive services, and social security programs that can help older persons avoid potential problems as they age. In general, improving the health of the older population would result in lower government spending on healthcare needs.

## 5. Conclusions

Differences in the prevalence of NCDs and preventive interventions can be tracked to aid in the development of effective health equity strategies. Our findings show that the wealthy have a higher prevalence of HTN and DM than the poor section, as well as significant disparities in preventive care among people with HTN or DM. Taken together, the disparities in the prevalence and access to preventive care may be both a cause and a result of socioeconomic disparities. According to the study, the most important socioeconomic variables to track inequality for availing the NCDs preventive care in India are economic level, education, employment, and status of residence. These findings have policy implications in terms of reducing economic disparities in HTN prevalence, for as by giving more labor opportunities to economically disadvantaged groups. Preventive care for NCDs should be considered an essential component of top-down and bottom-up public health policies. Advancement of NCD prevention strategies should be part of the broader public policy framework that includes the participation of both the public and private sectors. This will create a beneficial environment for all socioeconomic groups by improving the healthcare system fairly. Future research may investigate the causal relationships between the socioeconomic determinants of NCD and their inequality.

## Figures and Tables

**Figure 1 geriatrics-07-00137-f001:**
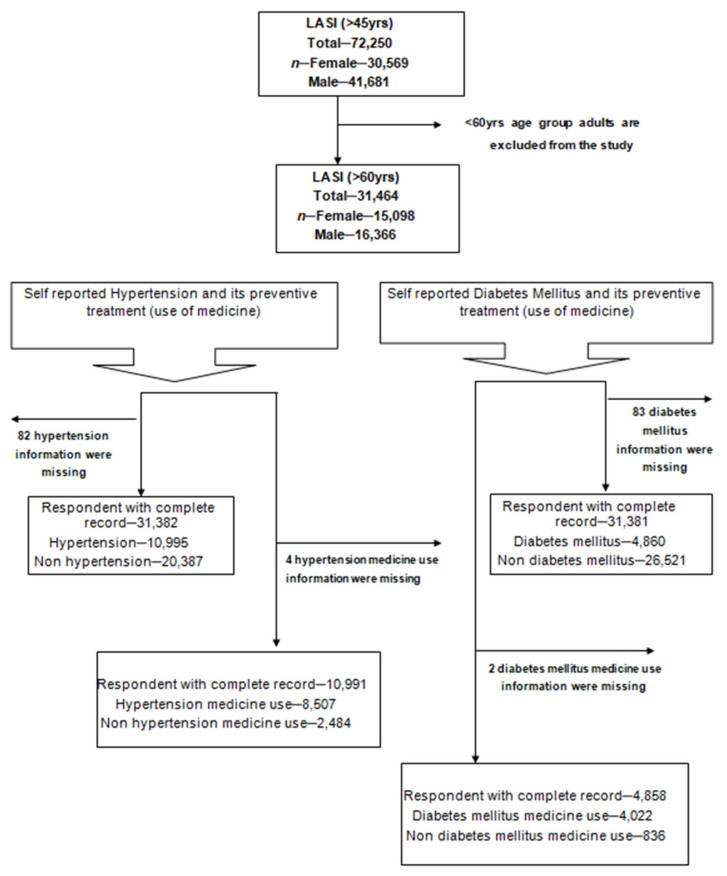
Schematic figure explaining sample extraction from LASI (2017–2018).

**Figure 2 geriatrics-07-00137-f002:**
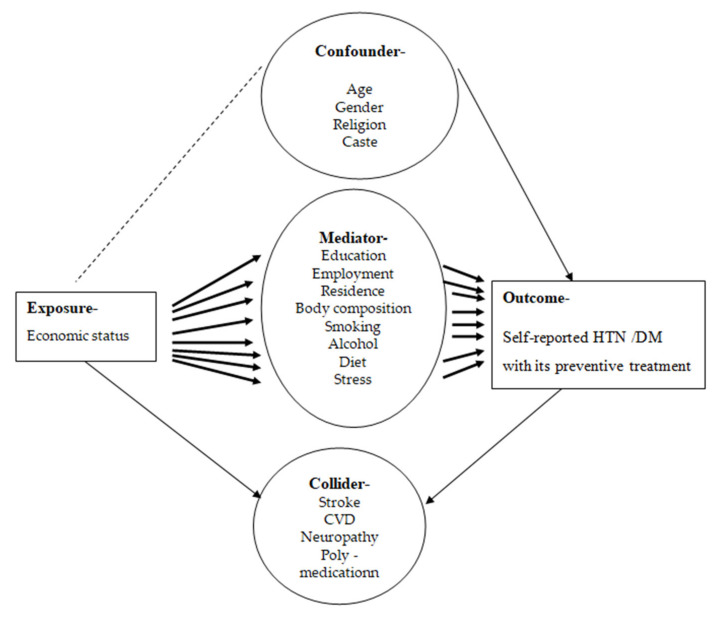
Simplified direct acyclic graph exhibiting the hypothesized causal association between economic status and self-reported HTN/DM and its preventative treatment.

**Figure 3 geriatrics-07-00137-f003:**
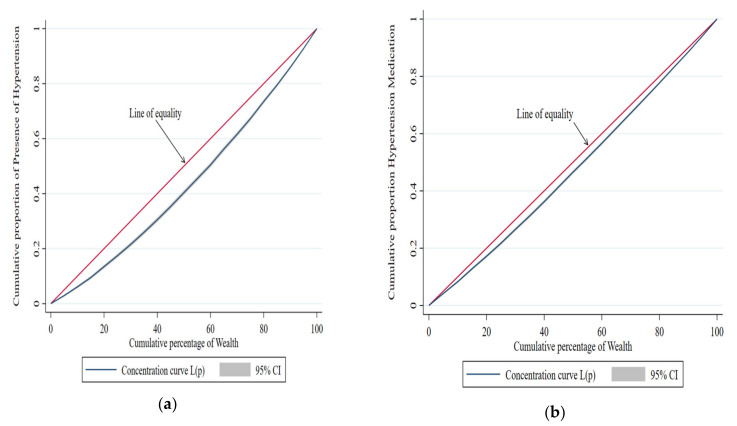
Concentration curve for the cumulative proportion of the presence of hypertension and the use of hypertension medications, against the cumulative percentage of wealth, India (LASI). (**a**) Description of the cumulative proportion of presence of hypertension in the first panel; (**b**) description of the use of the cumulative percentage of hypertension medications in the second panel.

**Figure 4 geriatrics-07-00137-f004:**
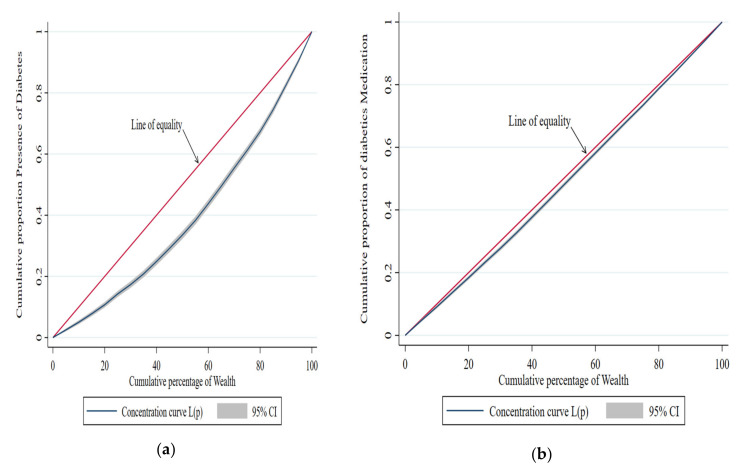
Concentration curve for the cumulative proportion of the presence of diabetes and the use of diabetes medicines, versus the cumulative percent of wealth, India (LASI). (**a**) Description of the cumulative proportion of presence of diabetes in the first panel; (**b**) Description of the cumulative proportion of presence of diabetes medicine use in the second panel.

**Table 1 geriatrics-07-00137-t001:** Distribution of NCD and its prevention (treatment-seeking) among the respondents aged 60 years and above.

Characteristics Profile of the Respondents Aged 60 and above Population (*n* = 31,464)			
Variables	*n*	Proportion	CI
HTN and its preventive care			
HTN	10,995	0.33	[0.32–0.34]
Currently on medication	8507	0.77 ^a^	[0.76–0.79]
Diabetics and its preventive care			
DM	4860	0.14	[0.13–0.15]
Currently on medicine	4022	0.82 ^b^	[0.79–0.84]
Demographic variables			
Age group			
60–69	18,974	0.59	[0.57–0.60]
70–79	9101	0.3	[0.29–0.31]
80 and above	3389	0.11	[0.11–0.12]
Place of residence			
Rural	20,725	0.71	[0.69–0.72]
Urban	10,739	0.29	[0.28–0.31]
Gender			
Male	15,098	0.47	[0.46–0.49]
Female	15,366	0.53	[0.51–0.54]
Cast group			
Schedule caste	5140	0.19	[0.19–0.20]
Schedule tribe	5173	0.08	[0.08–0.09]
Other backward class (OBC)	11,886	0.46	[0.45–0.48]
Others	8218	0.26	[0.25–0.27]
Economic status			
Poorest	6484	0.22	[0.21–0.23]
Poorer	6477	0.22	[0.21–0.23]
Middle	6416	0.21	[0.20–0.22]
Richer	6170	0.19	[0.18–0.20]
Richest	5917	0.16	[0.15–0.17]
Education level			
Illiterate	16,889	0.57	[0.55–0.58]
Primary or below	7560	0.23	[0.22–0.23]
Secondary	5560	0.17	[0.16–0.18]
college and above	1455	0.04	[0.04–0.05]
Marital status			
Married	19,920	0.62	[0.60–0.63]
Single	11,544	0.38	[0.37–0.40]
Religion			
Hindu	23,037	0.82	[0.81–0.83]
Muslim	3731	0.11	[0.10–0.12]
Christian	3150	0.03	[0.03–0.03]
Other	1546	0.04	[0.03–0.04]
Employment status			
Unemployed	9307	0.42	[0.41–0.43]
Employed	13,373	0.58	[0.57–0.59]

Note: (1) Numbers: unweighted, proportion: weighted; (2) **^a^** preventive care proportions among individuals with HTN; (3) **^b^** preventive care proportions among individuals with DM.

**Table 2 geriatrics-07-00137-t002:** Showsthe proportion of NCDs and preventive care (medicine use) by economic quintile, as well as economic inequality (%).

	HTN and Its Preventive Care		Diabetes Mellitus and Its Preventive Care	
	Presence of HTN	Currently on Medication	Presence of DM	Currently on Medication
Respondents (*n*)	31,464	10,995	31,464	4860
Poorest, % (95%CI) ^a^	5.66 [5.20–6.21]	12.01 [10.72–13.43]	2.16 [1.89–2.46]	11.47 [9.43–13.89]
Poorer, % (95%CI) ^a^	6.26 [5.81–6.74]	14.37 [12.94–15.92]	2.25 [2.0–2.52]	12.35 [10.33–14.71]
Middle, % (95%CI) ^a^	6.52 [5.97–7.12]	15.07 [13.87–16.36]	2.61 [2.30–2.95]	13.92 [11.72–16.47]
Richer, % (95%CI) ^a^	7.28 [6.47–8.18]	18.09 [16.27–20.05]	3.51 [2.80–4.39]	21.32 [18.15–24.88]
Richest, % (95%CI) ^a^	7.06 [6.27–7.95]	17.87 [16.03–19.87]	3.73 [2.97–4.66]	22.54 [19.10–26.40]
RII ^b^, (95%CI)	0.29 *** [0.28–0.30]	0.75 *** [0.73–0.76]	0.16 *** [0.15–0.17]	0.81 *** [0.79–0.83]
RII ^c^, (95%CI)	0.18 *** [0.17–0.20]	0.56 *** [0.53–0.60]	0.06 *** [0.05–0.07]	0.69 *** [0.64–0.76]
C, (95%CI)	0.13 *** [0.09–0.16]	0.10 *** [0.07–0.13]	0.10 *** [0.06–0.14]	0.09 *** [0.04–0.15]

Note: (1) 95% CI: 95% confidence interval, “C”: concentration index, “RII”: relative index of inequality; (2) ^a^ shows weighted unadjusted prevalence estimates; (3) ^b^ the RII values were established using Poisson regressions with robust variance and adjusted for gender and age; (4) ^c^: RII values established by Poisson regressions and adjusted for all other explanatory variables; (5) *** *p* < 0.001.

**Table 3 geriatrics-07-00137-t003:** Decomposition of the concentration index of chronic diseases and preventive care.

	HTN and Its Preventive Care						Diabetes Mellitus and Its Preventive Care					
	Presence of HTN			Currently on Medication			Presence of DM			Currently on Medication		
	dy/dx	Con.	%con.	dy/dx	Con.	%con.	dy/dx	Con.	%con.	dy/dx	Con.	%con.
Age Group (Ref: 60–69 years)												
70–79	0.018 *	<0.001	0.056	0.015	<0.001	0.06	−0.004	<0.001	−0.015	−0.001	<0.001	−0.005
80 and above	0.002	<0.001	−0.006	0.018	<0.001	−0.098	−0.031 ***	<0.001	0.168	−0.083 **	<0.001	0.48
Place of residence (Ref: rural)												
Urban	0.098 ***	<0.001	0.636	0.132 ***	0.001	1.091	0.086 ***	<0.001	0.726	0.076 ***	<0.001	0.675
Gender (Ref: female)												
Male	−0.086 ***	−0.005	−3.553	−0.042 **	−0.002	−1.99	−0.001	<0.001	−0.03	−0.062 **	−0.003	−3.211
Cast Group (Ref: schedule caste)												
Schedule tribe	−0.080 ***	0.002	1.303	−0.025	<0.001	0.527	−0.032 **	<0.001	0.681	−0.017	<0.001	0.373
Other backwardclass (OBC)	0.009	<0.001	0.099	0.047 ***	<0.001	0.671	0.016 **	<0.001	0.239	0	<0.001	0.005
None of them	0.023 *	0.003	2.571	0.051 **	0.007	7.304	0.012	0.002	1.81	0.015	0.002	2.381
Economic Status (Ref: poorest)												
Poorer	0.035 ***	−0.009	−7.089	0.065 ***	−0.017	−16.71	0.013	−0.003	−3.444	0.02	−0.005	−5.44
Middle	0.054 ***	0.003	2.268	0.072 ***	0.004	3.818	0.027 ***	0.001	1.471	0.012	<0.001	0.065
Richer	0.083 ***	0.023	17.787	0.108 ***	0.03	29.69	0.055 ***	0.015	15.41	0.115 ***	0.032	33.969
Richest	0.104 ***	0.037	28.61	0.143 ***	0.052	50.17	0.072 ***	0.026	25.85	0.107 ***	0.039	40.64
Education Level (Ref: illiterate)												
Primary or below	0.095 ***	0.002	1.627	0.054 ***	0.001	1.175	0.068 ***	0.001	1.52	0.057 **	0.001	1.35
Secondary	0.128 ***	0.009	7.443	0.066 ***	0.005	4.873	0.092 ***	0.007	6.917	0.050 *	0.004	3.994
college and above	0.133 ***	0.001	0.952	0.094 ***	0.001	0.856	0.088 ***	<0.001	0.816	−0.036	<0.001	−0.354
Marital Status (Ref: single)												
Married	−0.002	<0.001	−0.225	0.016	0.002	2.046	0.015 **	0.002	2.033	0.055 **	0.007	7.839
Religion (Ref: Hindu)												
Muslim	0.027 *	<0.001	−0.1	−0.018	<0.001	0.086	−0.014	<0.001	0.066	0.041	<0.001	−0.209
Christian	0.082 ***	<0.001	−0.01	0.101 ***	<0.001	−0.016	0.073 ***	<0.001	−0.012	0.05	<0.001	−0.009
Other	0.056 ***	<0.001	0.17	0.025	<0.001	0.097	0.015	<0.001	0.061	0.024	<0.001	0.099
Employment Status (Ref: unemployment)												
Employed	−0.108 ***	0.003	2.45	−0.090 ***	0.003	2.611	−0.060 ***	0.002	1.78	−0.076 ***	0.002	2.374

*p* * < 0.5, *p* ** < 0.01, *** *p* < 0.001.

**Table 4 geriatrics-07-00137-t004:** Description of factor contributions and age-sex-adjusted Cs.

	HTN and Its Preventive Care		Diabetes Mellitus and Its Preventive Care	
	Presence of HTN	Currently on Medication	Presence of DM	Currently on Medication
Age-sex groups, Con. (% con.)	−0.005 (−3.503%)	−0.002 (−2.028%)	<0.001 (0.123%)	−0.003 (−2.736%)
Economic status, Con. (% con.)	0.054 (41.576%)	0.069 (66.968%)	0.039 (39.287%)	0.066 (68.938%)
Other factors, Con. (% con.)	0.02 (16.916%)	0.02 (21.321%)	0.014 (16.637%)	0.016 (18.518%)
Residual, Con. (% con.)	0.061 (45.001)	0.013 (13.739%)	0.047 (43.953%)	0.011 (15.280%)
C	0.13	0.10	0.10	0.09
Age-sex adjusted C	0.135	0.102	0.10	0.093

Note: Con.: the absolute contribution of the determinants to the concentration index; % con.:the percentage contribution of determinants to the total concentration index; C: concentration index.

## Data Availability

The datasets generated and/or analyzed during this work are available in the repository of the International Institute for Population Sciences in Mumbai, India, and can be viewed freely in the public domain via http://iipsindia.org, accessed on 12 March 2021.

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
