# Peer review of "Socioeconomic Inequalities in the Prevalence of Non-Communicable Diseases among Older Adults in India"

_geriatrics, 2022, doi:10.3390/geriatrics7060137_

Round 1
Reviewer 1 Report
Khura et al. analysed data from the Longitudinal Ageing Survey of India (LASI) to investigate cross-sectionally the prevalence of non-communicable diseases (i.e., hypertension and diabetes) in older people (aged 60 years or more) across social strata, using a concentration index to assess socioeconomic inequality.
I have some specific comments that could help improve the paper’s quality.
- General English revision is needed throughout the manuscript.
- It could be useful to provide details on the eligibility criteria of the study population and the section of the participants.
- In the methods section, it could be advisable to add details on the efforts to address potential sources of bias and to explain how missing data were addressed.
- In the results section, please report the number of individuals at each stage of the study and consider using a flow diagram, also indicating the number of participants with missing data for each variable.
- The authors should consider adding some details on the weighting procedure they used.
- In the discussion, it could be advisable to add references to similar studies and available literature on the topic from other similar longitudinal studies on ageing from other high-income or low-middle-income countries in order to establish fruitful comparisons.
Author Response
We would like to thank the Editor and the esteemed reviewers for the careful and thorough reading of our paper and for the insightful comments and constructive suggestions.
Response to Reviewer 1
Comment: General English revision is needed throughout the manuscript.
Response: Thanks for your suggestion. We have made the necessary changes and revised the entire paper for clarity of language, grammar, and syntax.
Comment: It could be useful to provide details on the eligibility criteria of the study population and the section of the participants.
Response: Thanks for your valuable suggestion. We have elucidated the eligibility criteria of the study population and the section of the participants.
Comment: In the methods section, it could be advisable to add details on the efforts to address potential sources of bias and to explain how missing data were addressed.
Response: Missing data have been excluded from the analysis and some of the potential sources of bias were reduced by statistical adjustments and others come as limitations of the nature of data. A DAC has been included as an additional file in the manuscript
Comment: In the results section, please report the number of individuals at each stage of the study and consider using a flow diagram, also indicating the number of participants with missing data for each variable.
Response: As per your suggestion, we have incorporated theses changes in the revised manuscript.
Comment: The authors should consider adding some details on the weighting procedure they used.
Response: All estimates were computed using survey weights and accounting for the complex survey design.
Comment: In the discussion, it could be advisable to add references to similar studies and available literature on the topic from other similar longitudinal studies on aging from other high-income or low-middle-income countries in order to establish fruitful comparisons.
Response: As per your suggestion, necessary changes have been made.
Reviewer 2 Report
This submission by Khura, et al provides interesting results on prevention of hypertension and diabetes mellitus in a large population of older people from India. The study shows very demonstratively how much social inequalities impact the recognition, prevention, and treatment of these 2 diseases. The many factors analyzed , including educational level, marital status, employment, incomes make the study relevant in many aspects. Also, the parameters and methods the authors used, though hard to read by an unspecialized physician, are very finely designed to make the conclusions as reliable as required.
The large introduction section provides acute insights on the changes in the age-pyramid occuring in lower income countries, as well as the emerging occurrence of a 20% senior population in these countries. The methods section, though hard to read by medical doctors is nicely organized. The conclusions are clearly supported by the results the authors describe.
The main issue pertains to the many other publications on this topic, including several ones from India.
There is a lot of publications focusing on the impact of socioeconomic conditions on diabetes and hypertension, including among elderly. This has been reported in high-income countries, as well as in lower income countries, including several studies from India. See and add to the Reference Section :
1 Diabetes, elderly, socioeconomic : Richards SE, et al. PLoS One. 2022 Jul 28;17(7):e0270476
2 HBP :
Thrift AG, Hypertension in the rural India, the contribution of the socioeconomic position. J Am Heart Assoc 2020, e014486. doi: 10.1161/JAHA.119.014486.
Gatimu SM, et al : socioeconomic factors impact on HBP management in Kenya. Int J Equity Health. 2020;19:213. doi: 10.1186/s12939-020-01321-1.
In elderly :
- China : Xiao L, et al : BMC Cardiovasc Disord. 2021 ; 21:64.
- USA : Neufcourt, et al. J Hypertens 2021, 39 :2497 : Socioeconomic disparities and risk of hypertension among older Americans: the Health and Retirement Study
Taken this shortcomings into account, the approach the authors developed has never been used previously. Thats what make the study worthy and suitable for the interest of a very specialized readership of epidemiologists.
Author Response
We would like to thank the Editor and the esteemed reviewers for the careful and thorough reading of our paper and for the insightful comments and constructive suggestions.
Response to Reviewer 2
Comment: Taken this shortcoming into account, the approach the authors developed has never been used previously. That’s what makes the study worthy and suitable for the interest of a very specialized readership of epidemiologists.
Response: We sincerely appreciate your feedback and are grateful for your encouragement.